# Inhibition of defect-induced α-to-δ phase transition for efficient and stable formamidinium perovskite solar cells

Tian Chen[1,2], Jiangsheng Xie [1,2] ✉, Bin Wen[1,2], Qixin Yin[1,2], Ruohao Lin[1,2], Shengcai Zhu [1] ✉ & Pingqi Gao [1,2] ✉

Defects passivation is widely devoted to improving the performance of formamidinium lead triiodide perovskite solar cells; however, the effect of various defects on the α-phase stability is still unclear. Here, using density functional theory, we first reveal the degradation pathway of the formamidinium lead triiodide perovskite from α to δ phase and investigate the effect of various defects on the energy barrier of phase transition. The simulation results predict that iodine vacancies are most likely to trigger the degradation, since they obviously reduce the energy barrier of α-to-δ phase transition and have the lowest formation energies at the perovskite surface. A water-insoluble lead oxalate compact layer is introduced on the perovskite surface to largely suppress the α-phase collapse through hindering the iodine migration and volatilization. Furthermore, this strategy largely reduces the interfacial nonradiative recombination and boosts the efficiency of the solar cells to 25.39% (certified 24.92%). Unpackaged device can maintain 92% of its initial efficiency after operation at maximum power point under simulated air mass 1.5 G irradiation for 550 h.

The power conversion efficiency (PCE) of perovskite solar cells (PSCs) has reached a certified record value of 26%[1]. Since 2015, the state-of-the-art PSCs prefer to adopt formamidinium lead triiodide (FAPbI$_3$) perovskite as light absorption layer since it possesses the excellent thermal-stability and has a preferred bandgap closer to the Shockley-Queisser limit[2–4]. Unfortunately, the FAPbI$_3$ films thermodynamically undergo a phase transition from the black α-phase to a yellow non-perovskite δ-phase at room temperature[5,6]. Various complex perovskite compositions have been developed to prevent the formation of the δ-phase. The most common strategies to overcome this problem is mixing FAPbI$_3$ with a combination of methylammonium (MA$^+$), cesium (Cs$^+$) and bromide (Br$^-$) ions[7–9]. Nevertheless, mixed perovskites suffer an enlarged bandgap and light-induced phase segregation which will compromise the performance and operational stability of the resulting PSCs[10–12].

Recently, it is demonstrated that the pure FAPbI$_3$ single crystal without any alloying can own excellent stability due to its excellent crystallinity and low defects[13,14]. Thus, defects reduction through improving the crystallinity of bulk FAPbI$_3$ is an essential strategy for achieving efficient and stable PSCs[2,15]. However, degradation to undesirable yellow hexagonal non-perovskite δ-phase during the operation of FAPbI$_3$ PSCs can still occur[16]. This process is generally initialized from the surfaces and grain boundaries, which is more vulnerable under water, heat and light due to the exist of numerous defect sites[17]. Consequently, surface/grain boundary passivation is required to stabilize the black-phase FAPbI$_3$[18]. Many strategies of defects passivation, including introducing low dimensional perovskite, Lewis acid/base molecules and ammonium halide salts etc., have achieved great progress for the formamidinium PSCs[19–22]. To date, almost all studies focus on the role of various defects in determining the optoelectronic

[1]School of Materials, Shenzhen Campus of Sun Yat-sen University, No. 66, Gongchang Road, Shenzhen, Guangdong 518107, PR China. [2]Institute for Solar Energy Systems, State Key Laboratory of Optoelectronic Materials and Technologies, Sun Yat-sen University, Guangzhou 510275, PR China. ✉e-mail: xiejsh8@mail.sysu.edu.cn; zhushc@mail.sysu.edu.cn; gaopq3@mail.sysu.edu.cn

properties, such as carrier recombination, diffusion lengths and energy band structure in solar cells[22–24]. Such as using density functional theory (DFT), the formation energies and trap level of various defects are theoretically predicted, which has been widely used to guide the practical passivation design[20,25,26]. The device stability generally seems to improve as the defects decrease. However, the impact mechanism of various defects on the phase-stabilities and optoelectronic properties should be completely different in formamidinium PSCs. To our knowledge, a fundamental understanding of how the defects induce the cubic-to-hexagonal (α-to-δ) phase transition and the role of surface passivation on phase stability of α-FAPbI$_3$ perovskite are still not well-known.

Herein, by using DFT, we revealed the degradation pathway of the FAPbI$_3$ perovskite from black α-phase to yellow δ-phase and the effect of various defects on the energy barrier of α-to-δ phase transition. It is predicted that the I vacancies, which easily generated during the film preparation process and device operation, are most likely to trigger the α-to-δ phase transition. Thereby, we introduced a water-insoluble and chemically stable lead oxalate (PbC$_2$O$_4$) compact layer on the top of FAPbI$_3$ through in-situ reaction. The lead oxalate surface (LOS) suppressed the formation of I vacancies and retarded the migration of I ion under stimuli of heat, light and electric field. As a result, the LOS largely reduced the interfacial nonradiative recombination and boosted the efficiency of FAPbI$_3$ PSCs to 25.39% (certified 24.92%). Unpackaged LOS devices maintained 92% of their initial efficiency after operation at maximum power point (MPP) under simulated air mass (AM) 1.5 G irradiation for over 550 h.

## Results

### First-principles calculations studies

First, we performed first-principles calculations to find the degradation pathway of FAPbI$_3$ perovskite transition from α to δ-phase. From the detailed phase transition process, it is found that the transition of a 3D corner sharing [PbI$_6$] octahedron in cubic α-phase FAPbI$_3$ to a face-sharing 1D line [PbI$_6$] octahedron in hexagonal δ-phase FAPbI$_3$ is enabled by breaking 9 Pb–I bonds in the first step (Int-1) with energy barrier as high as 0.62 eV/cell, as shown in Fig. 1a. By the octahedrons shear along [0$\bar{1}$1] direction, the hexagonal short chains extend from 1 × 1 to 1 × 3, 1 × 4, and eventually transit to δ-phase. The orientation relation of the whole pathway is (011)$_\alpha$//(001)$_\delta$ + [100]$_\alpha$//[100]$_\delta$. While from the energy profile, we can find that after the δ-phase FAPbI$_3$ nucleation, the energy barrier become lower relative the α-phase in the following steps, meaning the phase transition will speed up. Obviously, if we want to inhibit the α-phase degradation, the manipulation of the first step of phase transition is critical.

Then, we systematically investigated the effect of various native point defects (including anti-site occupations of Pb$_{FA}$, I$_{FA}$, Pb$_I$, and I$_{Pb}$; interstitials of Pb$_i$, and I$_i$; vacancies of V$_I$, V$_{FA}$, and V$_{Pb}$), which were regarded as the key factor to trigger the phase degradation at atomic and energy level, as shown in Fig. 1b and Supplementary Table 1. Interestingly, not all the defects will reduce the energy barrier of α-to-δ phase transition (Fig. 1b). We believe that the defects which both have low formation energy and reduce the energy barrier of α-to-δ phase transition are considered to be harmful to the phase stability. As previous reports, Pb-rich surface is typically considered in efficient formamidinium PSCs[27]. Thus, we focused on the (100) surface with PbI$_2$ termination in a Pb-rich condition. The defect formation energies of native point defects on the surface are shown in Fig. 1c and Supplementary Table 1. Considering the energy barrier (EB) of phase transition and formation energy (FE), these defects were divided into three types. Type-I (Low EB-High FE): Although I$_{Pb}$, V$_{FA}$ and V$_{Pb}$ largely reduce the energy barrier of phase transition, they have high formation energies. Thus, we thought these types of defects have limited effect on the phase transition since they rarely formed. Type-II (High EB): The Pb$_I$, I$_{FA}$ and Pb$_{FA}$ antisite defects will not be detrimental to the phase

stability of α-FAPbI$_3$ perovskite due to the improved energy barrier of α-to-δ phase transition. Type-III (Low EB-Low FE): V$_I$, I$_i$ and Pb$_i$ defects with relatively low formation energies will induce the degradation of the black phase. Particularly lowest FE and EB of V$_I$ taken into consideration, we think that the most effective strategy is reducing the I vacancies.

In order to reduce the V$_I$, we designed a PbC$_2$O$_4$ compact layer to reinforce the FAPbI$_3$ surface. In comparison with the organic halide salt passivator, such as phenethylammonium iodide (PEAI) and n-Octylammonium Iodide (OAI), etc., PbC$_2$O$_4$ without mobile halogen ions is chemically stable and insoluble in water that can stabilize the perovskite surface well under the stimuli of moisture and electric field. The solubility of PbC$_2$O$_4$ in water is merely 0.00065 g liter$^{-1}$, which is even lower than that of PbSO$_4$[28]. More importantly, the compact and uniform LOS layer can be mildly prepared on the perovskite film through in-situ reaction (see below). We performed DFT simulation for the interface binding between FAPbI$_3$ and PbC$_2$O$_4$, as shown in Supplementary Fig. 1. The defect formation energies after introducing the LOS were given in Supplementary Table 2. We found that the LOS not only increases the V$_I$ defect formation energy from 0.69 eV to 1.53 eV (Fig. 1d), but also improves the activation energies for I migration on the surface and migration escaping the surface (Fig. 1e). For the first step, the I ion migrate on the perovskite surface leaving V$_I$ in the lattice site with an energy barrier of 0.61 eV; while the activation energy for I ionic migration increase to 1.28 eV after introducing LOS due to the steric-hindrance effect. For the I ionic migration escaping the perovskite surface, the energy barrier in LOS is also higher than that in control sample (Fig. 1e). The schematic views of I ionic migration path in the control and LOS FAPbI$_3$ are shown in Fig. 1f, g, respectively. The simulation results illustrate that the LOS can suppress the V$_I$ defects formation and I volatilization, which will in turn hinder the α-to-δ phase transition nucleation.

### Characterization of lead oxalate on the perovskite surface

The reaction between the oxalic acid and FAPbI$_3$ perovskite was tested. After mixing oxalic acid and FAPbI$_3$ solution, a large amount of white-colored sediment was yielded, as shown in Supplementary Fig. 2. The powder products were identified as pure PbC$_2$O$_4$ materials through X-ray diffraction (XRD) patterns (Supplementary Fig. 3) and fourier transform infrared spectroscopy (FTIR) (Supplementary Fig. 4). We found that oxalic acid can be dissolved in isopropanol (IPA) well at room temperature with a solubility of approximately 18 mg mL$^{-1}$, as shown in Supplementary Fig. 5. It is beneficial to the post-treatment, since the IPA as a general passivation solvent is not destructive to the perovskite layer in short time[29]. Thus, a thin and compact PbC$_2$O$_4$ can be rapidly generated on the surface of the perovskite films through immersing the perovskite film into the oxalic acid solution or spin-coating the oxalic acid solution on perovskite, according to the following chemical equations: H$_2$C$_2$O$_4$ + FAPbI$_3$ = PbC$_2$O$_4$ + FAI + HI. The FAI can be dissolved in the IPA and thus removed during the preparation process. The thickness of the LOS can be controlled by the reaction time and precursor concentration.

The scanning electron microscopies (SEM) of control and LOS perovskite films are shown in Fig. 2a, b. The results show that the perovskite surface morphology maintained well and a mass of small particles deposited on the surface of crystalline grain, which should be the PbC$_2$O$_4$ layer generated through in-situ reaction. The LOS perovskite films have a slight smoother surface (Supplementary Fig. 6) and a larger water contact angle than the control films (Supplementary Fig. 7). Cross-sectional high-resolution transmission electron microscopy (HR-TEM) was used to distinguish the surface product layers. As compared to the control film (Fig. 2c), a uniform and compact thin layer with a thickness of around 10 nm on the top of LOS perovskite could be clearly discerned (Fig. 2d). The high-angle annular dark field scanning transmission electron microscopy (HAADF-STEM) was used

to probe the interface between $PbC_2O_4$ and $FAPbI_3$ and it can be clearly observed that there is a crystalline $FAPbI_3$ region and an amorphous $PbC_2O_4$ region (Supplementary Fig. 8). X-ray photoemission spectroscopy (XPS) measurements were carried out to characterize the perovskite surface composition after oxalic acid treatment, as shown in Fig. 2e–g. In Fig. 2e, the C 1s peaks around 284.8 eV and 288.5 eV belonged to the specific signals of C−C and FA, respectively. Compared with the control film, an additional peak at 289.2 eV attributed to the $C_2O_4^{2-}$ was observed in the LOS film. In the O 1s spectra, the LOS perovskite exhibits three chemically distinct O 1s peaks at 531.7 eV, 532.5 eV, and 533.4 eV corresponding to the O atoms of deprotonated COO, C=O and OH components of the intact oxalate group[30] (Fig. 2f). For the control sample, only a slight O 1s peak is observed which can be attributed to the chemisorbed oxygen on the surface. The characteristic Pb $4f_{7/2}$ and Pb $4f_{5/2}$ were located at 138.4 eV and 143.3 eV for the control film, respectively. We observed that the LOS perovskite shows a -0.15 eV shift of the Pb peaks toward larger binding energies, which indicates the stronger interaction between $C_2O_4^{2-}$ and Pb atom (Fig. 2g).

## Stabilizing the pure α-phase $FAPbI_3$

Based on DFT results, $V_I$ defects and I migration are theoretically predicted to be liable to induce the α-to-δ phase transition. Previous reports have demonstrated that a fast release of $I_2$ from the FA-based perovskite films during light soaking, after exposing films under the stress of light and heat[31–33]. To confirm the lead oxalate stabilizing effect on the α-phase perovskite, we immersed the control and LOS perovskite films into transparent glass bottle filled with toluene, respectively, and then illuminated them under 1-sun illumination for 24 h. The ultraviolet-visible (UV−Vis) absorption of the toluene solution was measured, as shown in Fig. 3a. It is observed a much lower $I_2$ absorption intensity in the LOS perovskite case in comparison with the

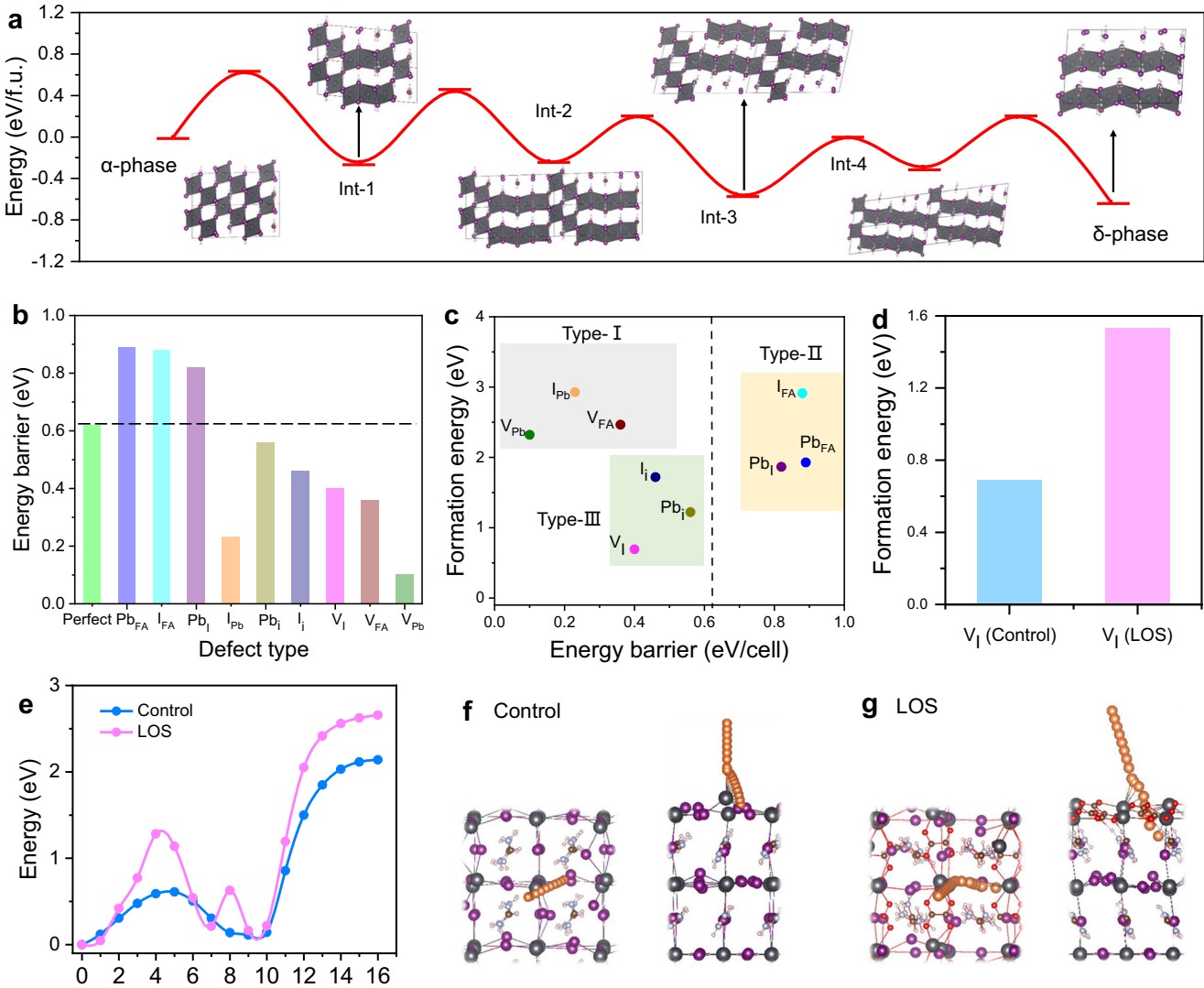

**Fig. 1 | First-principles calculations studies. a** The phase transformation process from left to right is black-phase $FAPbI_3$ (α-phase), the first interrupting of Pb−I bonds (Int-1), further breaking of Pb−I bonds (Int-2, Int-3, and Int-4), and yellow-phase $FAPbI_3$ (δ-phase). **b** The energy barrier of α-to-δ phase transition of $FAPbI_3$ based on various native point defects. The dashed line shows the energy barrier of perfect crystal (0.62 eV). **c** The defect formation energies of native point defects on the Pb-rich perovskite surface. The abscissa is energy barrier of α-to-δ phase transition and the ordinate is the formation energy of defects. The gray, yellow and green shaded parts are Type-I (Low EB-High FE), Type-II (High FE) and Type-III (Low EB-low FE), respectively. **d** The defect formation energies of $V_I$ in control and LOS $FAPbI_3$. **e** The I ion migration barrier in control and LOS $FAPbI_3$. **f, g** The schematic views (left: top views; right: cross-section views) of I ionic migration (orange balls) in **f** control and **g** LOS $FAPbI_3$ (gray, lead; purple (orange), iodine (mobile iodine); brown, carbon; light blue, nitrogen; red, oxygen; light pink, hydrogen). Source data are provided as a Source Data file.

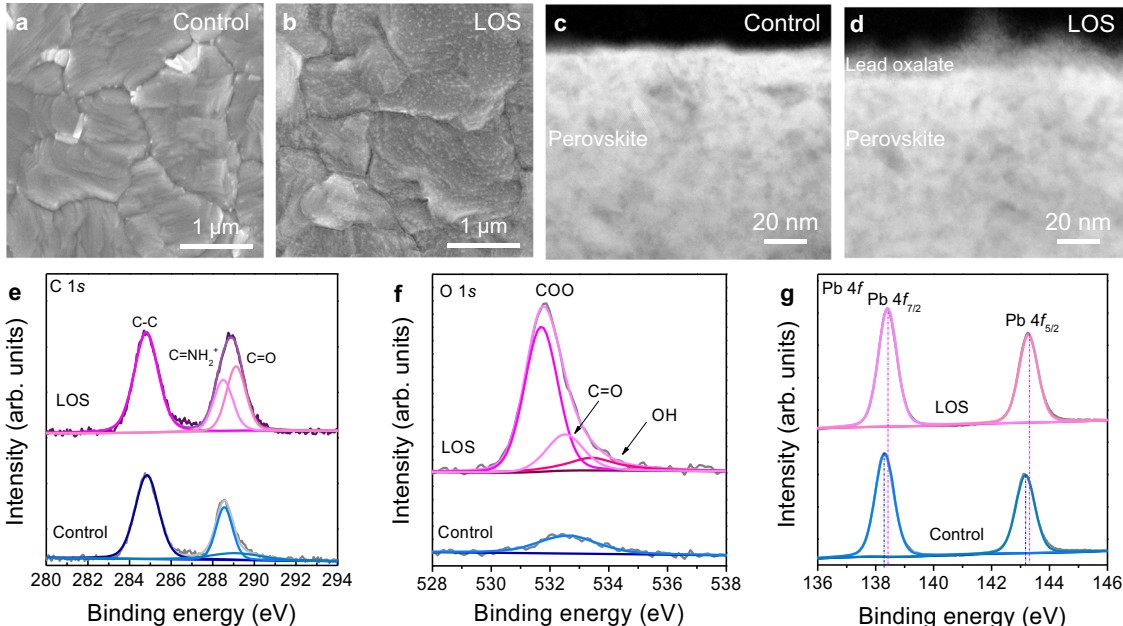

**Fig. 2 | Morphology and XPS spectra of control and LOS perovskite films.** Top-view SEM images of **a** control and **b** LOS perovskite films. Cross-sectional high-resolution transmission electron microscopy (HR-TEM) of **c** control and **d** LOS perovskite films. High-resolution XPS of **e** C 1*s*, **f** O 1*s* and **g** Pb 4*f* for the perovskite films. Source data are provided as a Source Data file.

control sample, illustrating that the compact LOS could suppress the release of $I_2$ from the perovskite films during light soaking. The photograph of the aged control and LOS perovskite films is shown in the inset of Fig. 3b and c. The LOS perovskite still maintained black while most region of the control film has already turned yellow. The UV–Vis absorption spectra of the immersed films were presented in Fig. 3b, c. We observed that the absorption corresponding to the α-phase in control film showed an obvious decrease. The XRD measurement was carried out to record the crystal structure evolution. A strong yellow δ-phase signal (11.8°) was observed in the control perovskite after 24 h illumination while the LOS perovskite still maintained the black-phase well (Fig. 3d).

We performed scanning electron microscopy (SEM) measurements to observe the microstructure changes of perovskite films after 24 h illumination, as shown in Fig. 3e, f. For the control film, the large crystal grains were damaged and converted to small needles which are consistent to the morphology of δ-phase $FAPbI_3$ product[17] (Fig. 3e). For the LOS film, the perovskite crystal grains maintained well (Fig. 3f). The results confirmed that the loss of I obviously induced the transition from black to yellow phase while the $PbC_2O_4$ stabilized the black phase through impeding the I loss. Owing to the much higher vacancy density at the surface than in the bulk of the grains[34], this phase transition is more likely to happen at the surface of the grains along with the release of iodine and the generation of $V_I$. As the DFT predicted, the LOS could suppress the formation of $V_I$ defects and block the I ions migration at the perovskite surface.

In addition, we studied the effect of $PbC_2O_4$ layers on the moisture resistance of perovskite films in ambient air (30–60% relative humidity). As presented in Supplementary Fig. 9, the LOS perovskite still remained black after 12 days, while the control film has already turned yellow. In XRD measurements, the control film exhibited strong peak at 11.8° corresponding to δ-phase $FAPbI_3$ while the LOS perovskite maintained black α-phase well (Supplementary Fig. 10).

### Carrier extraction and photovoltaic performance
The steady-state photoluminescence (PL) and time-resolved PL (TRPL) were performed to investigate the passivation effect of lead oxalate on the perovskite surface. Figure 4a shows the LOS film has an enhanced

PL intensity. In the PL mapping images, the intensity of LOS film in the whole $10 \times 10\ \mu m^2$ region was higher than that of control film (Supplementary Fig. 11), clarifying that the $PbC_2O_4$ uniformly passivates the perovskite film. The carrier lifetimes are determined through fitting the TRPL decays to a mono-exponential function (Fig. 4b). The LOS film shows a carrier lifetime of 5.2 μs which is much longer than the control film with a carrier lifetime of 0.9 μs, demonstrating the reduced nonradiative recombination at the surface.

To study the effect of lead oxalate layer on the device performance, a conventional n-i-p structure of $FTO/TiO_2/SnO_2$/perovskite/spiro-OMeTAD/Au was adopted. We used formamidinium chloride (FACl) as the additive instead of methylamine hydrochloride (MACl) in the perovskite precursor to achieve better device performance since FACl can result into better crystalline quality and avoid broadening the bandgap of $FAPbI_3$[35] (detailed comparisons are shown in Supplementary Figs. 12–14). The IPA was selected as an antisolvent since it resulted in better crystal quality and higher preferred orientation in perovskite films in comparison with the diethyl ether (DE) or chlorobenzene (CB)[36] (Supplementary Figs. 15 and 16). The thickness of $PbC_2O_4$ was carefully optimized through regulating the oxalic acid concentrations to well balance the defect passivation and charge transport (Supplementary Fig. 17). The cross-sectional SEM images of the post-optimized control and LOS devices are shown in Supplementary Fig. 18. The typical current density-voltage (*J-V*) curves of the control and LOS devices are shown in Fig. 4c, with the extracted parameters given in Supplementary Table 3. The control cell had a maximum power conversion efficiency (PCE) of 23.43% (22.94%) with a $J_{sc}$ of 25.75 mA cm$^{-2}$ (25.74 mA cm$^{-2}$), a $V_{oc}$ of 1.16 V (1.16 V) and a fill factor (FF) of 78.40% (76.69%) from reverse (forward) scan. The LOS PSC had a maximum PCE of 25.39% (24.79%) with a $J_{sc}$ of 25.77 mA cm$^{-2}$ (25.77 mA cm$^{-2}$), a $V_{oc}$ of 1.18 V (1.18 V) and an FF of 83.50% (81.52%) from reverse (forward) scan. The LOS device achieved a certified photovoltaic performance of 24.92% at a credible third-party photovoltaic laboratory (Supplementary Fig. 19). The external quantum efficiency (EQE) yielded an integrated $J_{sc}$ of 24.90 mA cm$^{-2}$ for the control and 25.18 mA cm$^{-2}$ for the LOS PSCs, respectively, which matched well with the measured $J_{sc}$ under AM 1.5 G standard

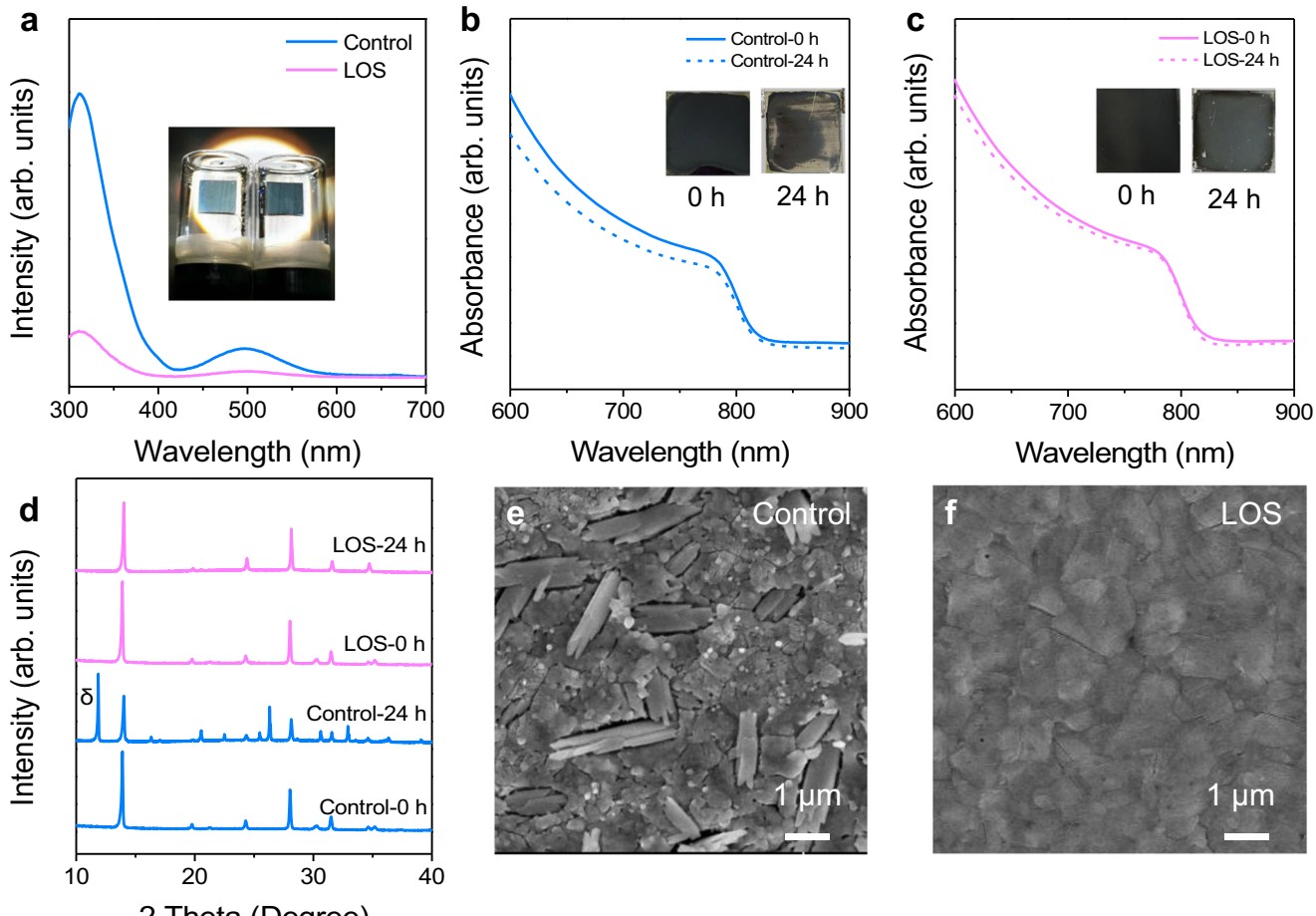

**Fig. 3 | Stability test of the FAPbI₃ films. a** UV–Vis absorption spectra of the toluene solutions in which control and LOS films were immersed under 1-sun illumination for 24 h. Inset is the picture of the vials in which each film was immersed into an equal volume of toluene. UV–Vis absorption spectra of **b** the control and **c** LOS films before and after immersed under 1-sun illumination for 24 h. Inset is the picture of the tested films. **d** XRD patterns of the control and LOS films before and after exposing for 24 h. SEM images of **e** the control and **f** LOS films after exposing for 24 h. Source data are provided as a Source Data file.

spectrum (Fig. 4d). A statistical distribution of the measured PCE of the control and LOS PSCs are shown in Supplementary Fig. 20.

As shown in Fig. 4e, the relationship between $V_{oc}$ and light intensity was calculated to investigate the effect of PbC₂O₄ on trap-assisted surface recombination. The slope of the fitting line for LOS device was 1.16 $k_B T/q$, which was lower than that of control device (1.31 $k_B T/q$), confirming that LOS was beneficial to suppress the trap-assisted surface recombination. We employed the space charge limited current (SCLC) technique to quantitatively measure the defect densities of the perovskite films through measuring the dark $I–V$ curves of hole-only devices (ITO/PEDOT:PSS/perovskite/spiro-OMe-TAD/Au), as shown in Fig. 4f. The trap densities were calculated through employing the formula of $N_t = 2\varepsilon_0 \varepsilon V_{TFL}/eL^2$, where $\varepsilon$ is the relative dielectric constant of perovskite film, $\varepsilon_0$ is the vacuum permittivity, $V_{TFL}$ is the trap filled limit voltage, $e$ is the charge, and $L$ is the thickness of the perovskite film (650 nm). The defect density of LOS device was calculated to be 1.450 × 10¹⁵ cm⁻³, which was lower than that of the control device with 1.795 × 10¹⁵ cm⁻³.

**Stability of PSCs**

The unencapsulated devices were tested at the maximum power point (MPP) under full-sun illumination in a nitrogen atmosphere to explore its long-term operational stability (Fig. 5a). The LOS device maintained 92% of its maximum efficiency after 550 h, while the control device degraded to 60% of its initial performance. The elemental distributions in the aged device were measured by time-of-flight secondary-ion mass

spectrometry (ToF-SIMS) measurement (Fig. 5b, c). Large accumulations of iodine can be seen for the control device along the top gold region. The inert gases protected condition ruled out environmental degradation factors, such as moisture and oxygen, indicating that intrinsic mechanisms (i.e., ion migration) were responsible. According to the ToF-SIMS results, I⁻ and AuI₂⁻ ions were found in the Au electrode, indicating that I diffused from perovskite to Au. Both the signal intensities of I⁻ and AuI₂⁻ ions in control device were approximately ten times higher than those of LOS sample. Previous report demonstrated that the ion penetration can lead to rapid degradation in hole conductivity of spiro-OMeTAD and chemical corrosion of the top electrode layer, which deteriorates the interface contact in the devices[37,38]. We peeled the Au electrode and washed the spiro-OMeTAD layer from the substrates by chlorobenzene solution. Then, we performed grazing incidence x-ray diffraction (GIXRD) characterization on the films (Fig. 5d). The results showed that an obvious diffraction peak located at 11.8° in the control film while there was no new peak in the LOS sample. It demonstrated that the massive loss of I ions from the control films induced the generation of δ-phase while this process was obviously suppressed in the LOS films.

The temperature dependent conductivity was measured to confirm that ion migration can be suppressed by the PbC₂O₄ (Supplementary Fig. 21). The activation energy ($E_a$) for the ion migration was determined by measuring the conductivity ($\sigma$) changes of FAPbI₃ films at different temperatures ($T$) with the Nernst–Einstein relation: $\sigma T = \sigma_0 \exp(-E_a/k_B T)$, where $\sigma_0$ is a constant, $k_B$ is the Boltzmann's constant.

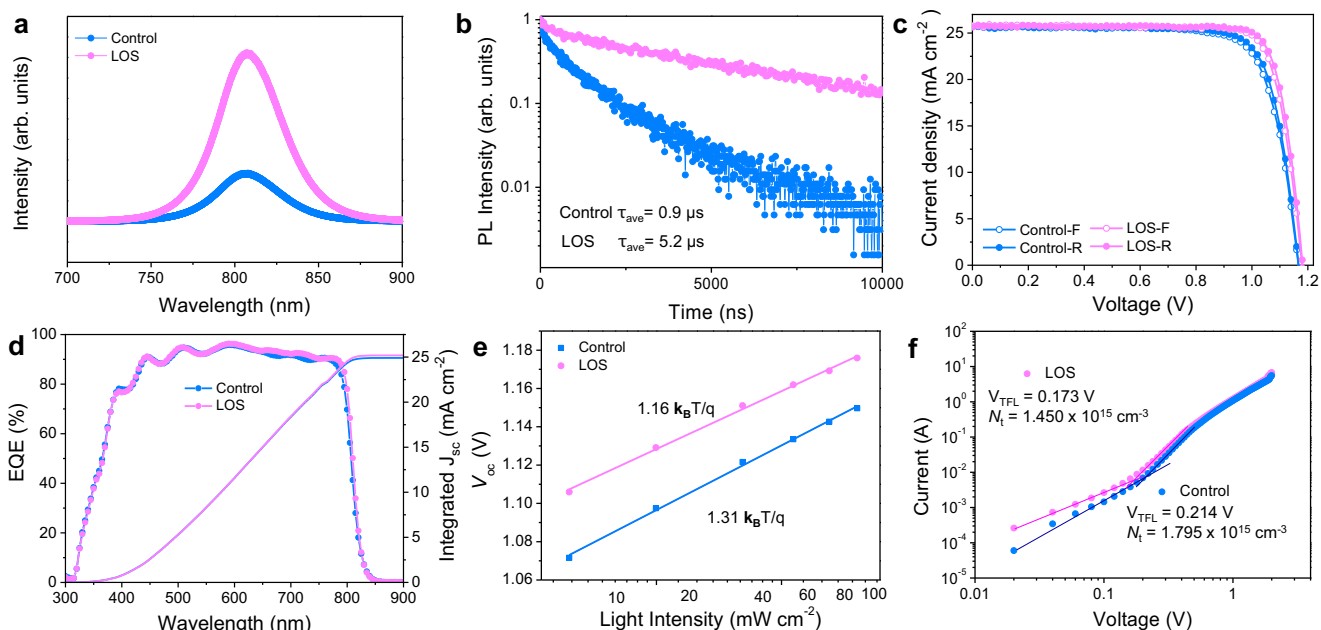

**Fig. 4 | Carrier extraction and photovoltaic characteristics.** PL of **a** steady-state PL and **b** time-resolved PL spectra of perovskite films on glass substrates. **c** $J$–$V$ curves of devices (FTO/TiO$_2$/SnO$_2$/perovskite/spiro-OMeTAD/Au). **d** EQE spectrum and $J_{sc}$ integrated from the EQE spectrum of the best-performing devices. **e** Light-intensity dependence on $V_{oc}$ plots for the perovskite devices. **f** The typical SCLC analysis with the hole-only device of ITO/PEDOT:PSS/perovskite/PCBM/Au. $V_{TFL}$ is the trap filled limit voltage. From the data we calculated the trap densities ($N_t$). Source data are provided as a Source Data file.

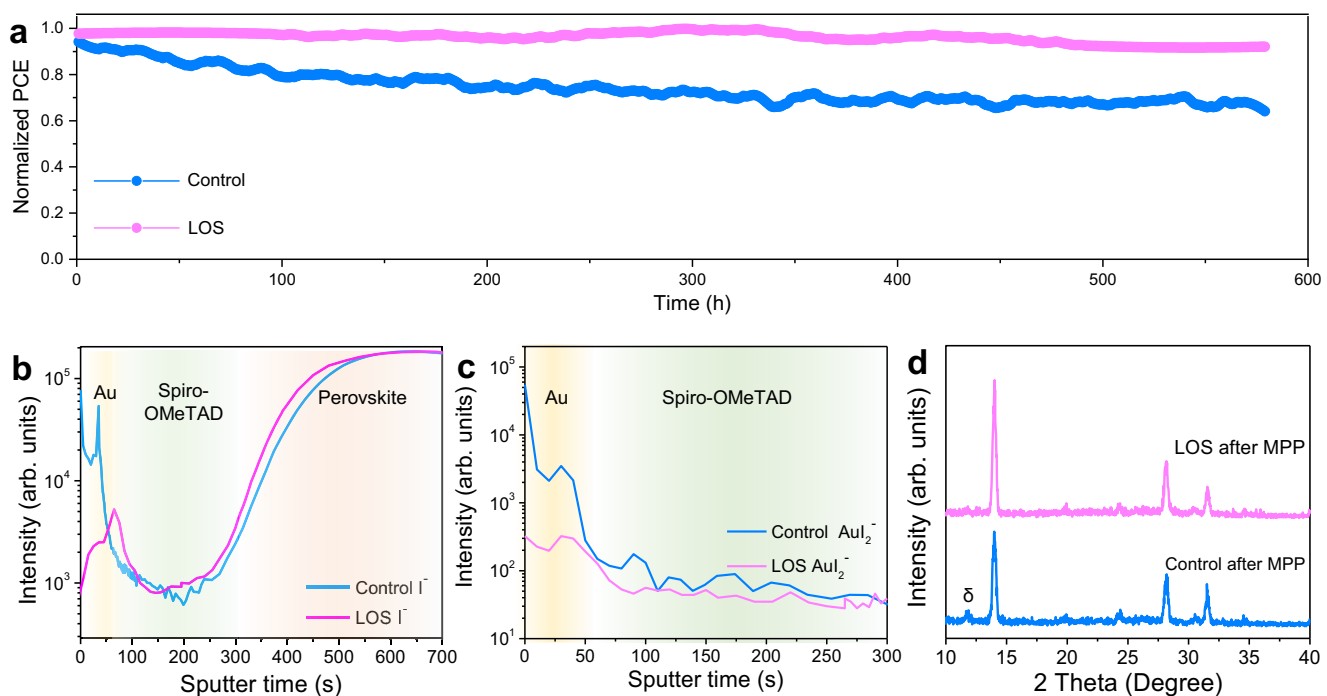

**Fig. 5 | Stability of control and LOS PSCs. a** Continuous MPP tracking for 575 h of unsealed devices in a nitrogen atmosphere under 1-sun illumination without UV filter. ToF-SIMS of the **b** I$^-$ and **c** AuI$_2^-$ ions distributions in the control and LOS MPP-aged devices. The yellow, green and orange shadings corresponding to Au, Spiro-OMeTAD and perovskite. **d** GIXRD of the perovskite films after MPP test. Source data are provided as a Source Data file.

We derived $E_a$ values from the slope of the ln($\sigma$T) versus $1/T$ of ~0.283 eV for the control and 0.419 eV for the LOS device.

## Discussion
In summary, we give the theoretical framework for revealing the degradation pathway of the FAPbI$_3$ perovskite and the effect of various defects on the energy barrier of α-to-δ phase transition. Among these defects, $V_I$ defects are theoretically predicted to be liable to induce the α-to-δ phase transition. The water-insoluble and chemically stable PbC$_2$O$_4$ compact layer was introduced to stabilize the α-phase FAPbI$_3$ through suppressing the formation of I vacancies and migration of I ion. The strategy largely reduced the interfacial nonradiative

recombination and boosted the efficiency of the solar cells to 25.39% with improved operation stability. Our results provide the guideline for achieving efficient and stable formamidinium PSCs through inhibiting the defect-induced α-to-δ phase transition.

## Methods

### Materials

Titanium(IV) isopropoxide (TTIP, 99.999%) was purchased from Sigma-Aldrich. Hydrochloric acid (HCl, 35.0–37.0%) and ethanol (anhydrous) were purchased from Guangzhou Chemicals. The $SnO_2$ (15 wt% colloidal dispersion tin(IV) oxide) was purchased from Alfa Aesar. Lead (II) iodide ($PbI_2$, 99.99%) was purchased from TCI Shanghai (China). Formamidinium iodide (FAI, ≥99.5%), Formamidinium Chloride (FACl, ≥99.5%), methylamine hydrochloride (MACl, ≥99.5%), 2,2′,7,7′-tetrakis-(N, N-di-p-methoxyphenylamine)−9,9′spirobifluorene (Spiro-OMeTAD, ≥99.5%), bis(trifluoromethane) sulfon imide lithium salt (Li-TFSI, 99.95%), 4-tertbutylpyridine (*t*BP, 96%) were purchased from Xi'an Polymer Light Technology Corp. (China). N,Ndimethylformamide (DMF, 99.8%), dimethylsulfoxide (DMSO, 99.9%), isopropanol (IPA, 99.8%), chlorobenzene (CB, 99.8%), acetonitrile (ACN) were purchased from Sigma-Aldrich. Oxalic acid ($H_2C_2O_4$, 99.9%) was purchased from Macklin. All chemicals were used as received without any other refinement.

### Device fabrication

The ITO or FTO substrates (1.5 × 1.5 cm²) were cleaned ultrasonically with detergent, acetone, and ethanol for 10 min, respectively, then drying under nitrogen flow. A dense blocking layer of $TiO_2$ was deposited onto FTO substrate using titanium diisopropoxide bis(acetylacetonate) in ethanol (1/25, v/v) at 500 °C for 60 min. The $SnO_2$ colloidal dispersion was diluted with deionized water in a volume ratio of 1:5. The clean substrate that treated under UV-Ozone for 20 min was spin-coated with a thin layer of $SnO_2$ nanoparticle film at 4000 rpm for 30 s, then it was preheated at 150 °C for 30 min. For the perovskite precursor solution, 275.2 mg FAI, 737.6 mg $PbI_2$, and FACl (20% mol) were dissolved in an DMF/DMSO (15/1) mixed solvent. The perovskite layer was fabricated by spin-coating 40 μL perovskite precursor solution on top of the UV-Ozone treated $SnO_2$ layer at a speed of 5000 rpm for 25 s in ambient air. At 5 s from the last, 50 μL MACl IPA solution (4 mg mL⁻¹) as an antisolvent was rapidly dropped onto the substrate. Then the as fabricated film was annealed at 150 °C for 20 min, and followed by annealing at 100 °C for 10 min. After the perovskite film was cooled down to room temperature, $H_2C_2O_4$ solution (1, 2, 4 mg in 1 mL IPA) was spin-coated at 4000 rpm for 30 s for passivating perovskite surface. The spiro-OMeTAD solution prepared by mixing 72.3 mg spiro-OMeTAD, 1 mL CB, 27 μL *t*BP and 17.5 μL Li-TFSI (520 mg in 1 mL acetonitrile) was spin-coated on the films at 4000 rpm for 30 s. Finally, 100 nm Au layer was evaporated under vacuum at rates of 0.05 nm s⁻¹ (0-1 nm), 0.1 nm s⁻¹ (2-15 nm) and 0.5 nm s⁻¹ (16-100 nm).

### Device characterization

The $J–V$ characteristics of perovskite solar cells were measured using a Keithley 2400 meter under the illumination of the solar simulator (SS-X50) at a light intensity of 100 mW cm⁻² as checked with a calibrated standard silicon solar cell. Unless otherwise stated, the $J–V$ curves were all measured in a nitrogen-filled glovebox with forward and reverse scan modes (voltage steps of 20 mV and a delay time of 10 ms) at room temperature (~25 °C). A shadow mask was used to define an effective area of 0.067 cm² for the measured PSCs. EQE measurements were performed in ambient air using a PVE300-IVT210 system (Industrial Vision Technology (s) Pte Ltd) with monochromatic light focused on a device. For the device stability, the test of the non-encapsulated solar cell was carried in a nitrogen glovebox under 100 mW cm⁻² without UV filter. The ToF-SIMS was measured by PHI nanoTOFII Time-of-Flight SIMS. The depth profiling was obtained through a 4 KV Ar ion gun of 400 × 400 μm area.

### Film characterization

X-ray photoelectron spectroscopy (XPS) measurement was carried out on a Thermo-VG Scientific (ESCALAB 250) system with a monochromatized Al Kα (for XPS mode) under a pressure of 5.0 × 10⁻⁷ Pa. Scanning electron microscopy (SEM) was performed on a JEOL-JSM-6330F system. The surface morphology and roughness of the perovskite film were collected by atomic force microscope (AFM) (Bruker Dimension FastScan). STEM and HAADF-STEM was performed on FEI titan Themis STEM. UV-Vis absorption spectra were measured by UV-3600Plus (Shimadzu). Space charge limited current (SCLC) was recorded on Keithley 2400 meter. Steady-state photoluminescence (PL) and time-resolved photoluminescence (TRPL) decay for carrier lifetime were measured via a Photoluminescence Spectrometer FLS 1000. PL mapping images were measured by Horiba LabRam HR Evolution Raman system. Fourier transform infrared spectroscopy (FTIR) was performed by Thermo-Fisher Nicolet NXR 9650 system.

### Pathway sampling

In this work, we utilized the SSW pathway sampling method to explore the phase transition pathway of α-phase to δ-phase. In the SSW method, the movement on the potential surface is guided by the random soft mode (second derivative) direction, which is capable to explore potential energy surface exhaustively and unbiasedly. In this work, the pathway sampling is carried out in a 72-atom supercell, and more than 100 pairs of initial state /final state (IS/FS) are collected in DFT level. Based on the IS/FS pairs datasets, the pathways connecting the initial structure and the final structure can be determined in an atom-to-atom correspondence, then the transition state can be seamlessly located by using the variable-cell double-ended surface walking approach (VC-DESW). After transition state searching, the lowest energy barrier pathways can be determined by sorting the energy barriers.

### DFT calculations

All DFT calculations were performed using *VASP* (5.3.5 version), where the electron-ion interaction of C, N, H, Pb and I atoms were represented by the projector augmented wave (PAW) scheme. The exchange-correlation functional was described by the generalized gradient approximation in the Perdew−Burke−Ernzerhof parametrization[39]. The energy cutoff of the plane wave was set to 400 eV. The Monkhorst− Pack k-point mesh is set to (2 × 2 × 1). For all the structures, both lattice and atomic positions were fully optimized until the maximal stress component is below 0.1 GPa and the maximal force component below 0.02 eV/Å. In the surface model, the $FAPbI_3$ surface are 4 layers, the bottom one-layer atom fixed which mimics the bulk of $FAPbI_3$ while the top three-layer can move freely during the optimization. The $PbC_2O_4$ layer is 1 ML on the I-terminal $FAPbI_3$ surface with the Pb bonded with one I and four O.

### Reporting summary

Further information on research design is available in the Nature Portfolio Reporting Summary linked to this article.

## Data availability

All data generated or analyzed during this study are included in the published article and its Supplementary Information and Source Data files. The source data presented in this study is available at https://doi.org/10.6084/m9.figshare.24100164[40]. Source data are provided with this paper.

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

## Acknowledgements

The authors would like to thank Shenzhen Fundamental Research Program (Grant No. JCYJ20200109142425294, P.G), the National Natural Science Foundation of China (Grant No.22005354, J.X.), and Guangdong Basic and Applied Basic Research Foundation (Grant No. 2019A1515110905, J.X.) for financial support. Fundamental Research Funds for the Central Universities, Sun Yat-sen University (Grant Nos. 22qntd0205 and 23qnpy04, J.X. and S.Z.). We also acknowledge the use of computing resources from the Tianhe-2 Supercomputer.

## Author contributions

J.X. and P.G. supervised the work. J.X. and T.C. conducted the idea and designed the experiments, and prepared the manuscript. S.Z. conducted the DFT calculations. T.C. conducted most of the experiments. B.W., Q.Y., and R.L. assisted with experiments and device fabrication. P.G. contributed to the revision of the manuscript.

## Competing interests

The authors declare no competing interests.
