## [Peer Review File · Nature Communications]

Inhibition of defect-induced α -to- δ phase transition for efficient and stable formamidinium perovskite solar cellsREVIEWER COMMENTS

Reviewer #1 (Remarks to the Author):

See Attachment

This communication reveals the effect of various defects on inducing the α -to- δ phase transition in FAPbI₃, which are not well known before. The influence law of defects on the phase-stabilities and optoelectronic properties are quite different. They found the V_I has both the low formation energies and energy barrier of α -to- δ phase transition, which is most likely to lead to the FAPbI₃ degradation. A stable PbC₂O₄ surface was prepared to restrain the mobile I to stabilize the α phase. The champion PSCs efficiency reached 25.4% with an independent certified 24.92%. The results are exciting and timely, and conclusion is supported by experimental results. Before it can be acceptable in Nature Communication, a minor revision is needed:

- (1) This work fabricates FAPbI₃ by one-step method. We noted that the FAcI instead of the common MAcI was used as the additive to deposit the FAPbI₃ films. Is there any advantage for this component?
- (2) It is better to give the cross-sectional scanning electron microscopy image of the devices with and without passivation.
- (3) Line 161. The author found the PbC₂O₄ is “around 10 nm on the top of LOS perovskite”, while during the DFT calculation, they only using one PbC₂O₄. How to interpret the discrepancy of between theory and experiment?
- (4) In addition, some details should be added: The defect formation energies of native point defects on the Pb-rich perovskite surface (Figure 1c) should be clearly listed in the Table. The material information of H₂C₂O₄ should be added at “Materials”.

Reviewer #2 (Remarks to the Author):

The α - δ phase transition is an important factor that affects the stability of FA-based perovskite solar cells. In the manuscript, DFT was used to reveal the degradation pathway of the α - δ phase transition of FAPbI₃. PbC₂O₄ was introduced to suppress the transition and the target FAPbI₃ solar cells delivered a certified efficiency of 24.92%. Moreover, the device stability at MPP was enhanced. The work provides some interesting insights on the defects inducing α - δ phase transition. However, the investigation of the passivation mechanism is not in-depth and there are some issues needed to be clarified before it can be accepted.

(1) In the first-principles calculations, how does the PbC₂O₄ layer bind with FAPbI₃? The binding picture from the atomic level should be provided. In addition, experimental evidence such as HADDF is suggested to confirm the interface binding.

(2) According to the calculation, reducing the I vacancies is the most effective strategy for suppressing the α - δ phase transition. However, many substances have been reported to significantly reduce the I vacancies and what's the advantage of PbC₂O₄?

(3) It is difficult to observe the crystal lattice from Figure 2c,d.

(4) Are there any peaks corresponding to PbC₂O₄ in the XRD pattern of LOS perovskite films?

Reviewer #3 (Remarks to the Author):

This paper reported the pathway of α -to- δ phase and the effect of various defects on the phase-transition degradation of FAPbI₃ perovskite. They found that among the defects, the I vacancies easily generated on the surface and decreased the energy barrier of phase-transition, thus they are most likely to induce the α -FAPbI₃ degradation. Followed the DFT theoretical framework, a water-insoluble and stable lead oxalate was designed to stabilize the α -phase due to reduce the I vacancies and I migration. The resultant devices show very high efficiency and robust soaking stability. It is worthy publication in Nature Communications. Before it could be accepted, several points need to be clarified:

(1) There is lacking of a clear description on the interaction between H₂C₂O₄ and FAPbI₃. How about the uniformity of PbC₂O₄ layer on the perovskite? The PL mappings of the control and target films are also indispensable.

(2) The effect of bare IPA on the device performance should be studied to excluding the role of IPA.

(3) The IPA was rarely used as an anti-solvent to achieve high efficiency PSC. Most used diethyl ether or CB in one-step method. More detail about this method is needed.

(4) Besides VI, how about the changes of the other defect formation energies after introducing the LOS?

(5) Some English need correction. Such as Line 38 “various composition..... developed.” Line 72 “which are easily generated”, Line 168 “the LOS perovskite exhibited to three chemically”.

Responses to Reviewer

We sincerely thank the reviewer for the constructive and positive comments and suggestions to improve our manuscript. We have addressed all issues below in detail and made appropriate changes in the **Revised Manuscript** and **Revised Supplementary Information**. Please see the following responses to your questions raised.

*Original referee comments are reproduced in black; our responses are marked in blue and corrections are highlighted by a yellow background here and in **Revised Manuscript** as well as **Revised Supplementary Information**.*

Reviewer #1 (Remarks to the Author):

This communication reveals the effect of various defects on inducing the α -to- δ phase transition in FAPbI₃, which are not well known before. The influence law of defects on the phase-stabilities and optoelectronic properties are quite different. They found the V_I has both the low formation energies and energy barrier of α -to- δ phase transition, which is most likely to lead to the FAPbI₃ degradation. A stable PbC₂O₄ surface was prepared to restrain the mobile I to stabilize the α phase. The champion PSCs efficiency reached 25.4% with an independent certified 24.92%. The results are exciting and timely, and conclusion is supported by experimental results. Before it can be acceptable in Nature Communication, a minor revision is needed:

General response: We sincerely thank the reviewer's positive comments and we have comprehensively revised the manuscript according to the reviewer's valuable suggestions. Please see the following responses.

(1) This work fabricates FAPbI₃ by one-step method. We noted that the FAcI instead of the common MACI was used as the additive to deposit the FAPbI₃ films. Is there any advantage for this component?

Response: Thanks for the comment. In fact, N. G Park *et al.* have demonstrated that the FAcI additive has benefits over MACI because it will not change the bandgap of FAPbI₃ (Sol. RRL, 2020, 4, 2000331). In this work, we repeated their experiments for pursuing better base-line of PSCs performance. It was found that the FAcI additive resulted in a much larger grain of perovskite films than the MACI additive (Supplementary Fig. 12). As shown in UV-Vis spectra, the results showed that the FAcI perovskite has a narrower bandgap than MACI perovskite (Supplementary Fig. 13), which is consistent with the results of N. G Park *et al.* We further investigated the dependence of photovoltaic performance on the additives of FAcI and MACI in FAPbI₃-based perovskite solar cells (PSCs). From the typical *J-V* in Supplementary Fig. 14, as compared with the device with MACI-additive, the J_{sc} is improved for FAcI additive device, which

is consistent with the previous reports (Sol. RRL, 2020, 4, 2000331). Thus, we believe that the FACL is a better additive for PSCs performance.

Revision in manuscript:

We used formamidinium chloride (FACL) as the additive instead of methylamine hydrochloride (MACl) in the perovskite precursor to achieve better device performance since FACL can result into better crystalline quality and avoid broadening the bandgap of FAPbI_3 ³⁶ (detailed comparisons are shown in Supplementary Figs. 12-14).

Supplementary Figure 12. SEM images of the perovskite films with (a) FACL and (b) MACl as the perovskite precursor additive. The FACL perovskite film has larger grains than the MACl film.

Supplementary Figure 13. UV-vis absorption spectra of the perovskite films with FACL and MACl additive, respectively. The FACL perovskite film showed higher absorbance than MACl perovskite and broaden absorption band edge.

Supplementary Figure 14. Typical J - V curves of PSCs using FACL and MACl as the perovskite precursor additive, respectively.

(2) *It is better to give the cross-sectional scanning electron microscopy image of the devices with and without passivation.*

Response: Thanks very much for the suggestion. We have given the cross-sectional scanning electron microscopy images of the control (without passivation) and LOS devices (with passivation), as shown in “Supplementary Fig. 18”.

Revision in manuscript:

The cross-sectional SEM images of the post-optimized control and LOS devices are shown in Supplementary Fig.18.

Supplementary Figure 18. The cross-sectional SEM images of the (a) control and (b) LOS devices.

(3) Line 161. The author found the PbC_2O_4 is “around 10 nm on the top of LOS perovskite”, while during the DFT calculation, they only using one PbC_2O_4 . How to interpret the discrepancy of between theory and experiment?

Response: Thanks for the comment. Though the PbC_2O_4 layer is around 10 nm, only the first layer PbC_2O_4 on perovskite surface is chemical adsorption, more other PbC_2O_4 form on the first layer PbC_2O_4 with the amorphous state. Thus, to simulate the 10 nm amorphous PbC_2O_4 is beyond the state-of-art of DFT calculation ability, while it is more suitable to simplify the model to one-layer PbC_2O_4 on perovskite surface. The previous reports generally adopted the similar simplification (Science, 2019, 365, 473-478). Besides, from our simulation, we can find that the one-layer PbC_2O_4 can play a key role to the phase degradation kinetic in theory. A thicker PbC_2O_4 layer will protect the FAPbI_3 perovskite more effectively in the device in experiment.

Revision in manuscript:

In the surface model, the FAPbI_3 surface are 4 layers, the bottom one-layer atom fixed which mimics the bulk of FAPbI_3 while the top three-layer can move freely during the optimization. The PbC_2O_4 layer is 1 ML on the I-terminal FAPbI_3 surface with the Pb bonded with one I and four O.

(4) In addition, some details should be added: The defect formation energies of native point defects on the Pb-rich perovskite surface (Figure 1c) should be clearly listed in the Table. The material information of $\text{H}_2\text{C}_2\text{O}_4$ should be added at “Materials”.

Response: Thanks for your suggestions, we have listed the defect formation energies of native point defects (Fig. 1c) in the “Supplementary Table 1” and added the material information of $\text{H}_2\text{C}_2\text{O}_4$ at “Materials” as follow.

Revision in manuscript:

Supplementary Table 1. The energy barrier of phase transition and formation energies of native point defects on the surface.

Type	Type I			Type II			Type III		
Defect	V _{Pb}	V _{FA}	I _{Pb}	Pb _I	Pb _{FA}	I _{FA}	V _I	I _i	Pb _i
Energy barrier of phase transition (eV)	0.1	0.36	0.23	0.82	0.89	0.88	0.4	0.46	0.56
Formation energy (eV)	2.32	2.46	2.93	1.87	1.93	2.91	0.69	1.72	1.22

Materials: Oxalic acid (H₂C₂O₄, 99.9%) was purchased from Macklin.

Reviewer #2 (Remarks to the Author):

The α - δ phase transition is an important factor that affects the stability of FA-based perovskite solar cells. In the manuscript, DFT was used to reveal the degradation pathway of the α - δ phase transition of FAPbI₃.PbC₂O₄ was introduced to suppress the transition and the target FAPbI₃ solar cells delivered a certified efficiency of 24.92%. Moreover, the device stability at MPP was enhanced. The work provides some interesting insights on the defects inducing α - δ phase transition. However, the investigation of the passivation mechanism is not in-depth and there are some issues needed to be clarified before it can be accepted.

General response: We sincerely thank the reviewer for reviewing the manuscript and giving valuable comments. It is really helpful to improve this work. In the revised manuscript, we give more evidences to clarify the passivation mechanism.

(1) In the first-principles calculations, how does the PbC₂O₄ layer bind with FAPbI₃? The binding picture from the atomic level should be provided. In addition, experimental evidence such as HAADF is suggested to confirm the interface binding.

Response: Thanks for your valuable suggestions. The binding picture of PbC₂O₄ layer on FAPbI₃ surface in our DFT calculation is added to Supplementary Fig. 1. In the surface model, the FAPbI₃ surface are 4 layers, the bottom one-layer atom fixed which mimics the bulk of FAPbI₃ while the top three-layer can move freely during the optimization. The PbC₂O₄ layer is 1 ML on the I-terminal FAPbI₃ surface with the Pb bonded with one I (from FAPbI₃ surface) and four O (from C₂O₄²⁻).

As shown in Supplementary Fig. 8c and d, high-angle annular dark-field (HAADF) STEM has been used to observe the interface between PbC₂O₄ and FAPbI₃ at atomic level. A pronounced perovskite lattice can be observed and the crystalline region with an interplanar spacings of 0.32 nm was identified as

the (200) domain of α -FAPbI₃ (Supplementary Fig. 8d) (Nature Energy, 2021, 6, 419-428. Science 2023, 379, 288–294). There was an amorphous region in another side, which should be the PbC₂O₄. The boundary between FAPbI₃ and PbC₂O₄ can be observed. However, the interface binding at atomic level is not clear since the PbC₂O₄ layer is amorphous. Similar phenomenon was found in previous reports (Science 365, 473–478 (2019)).

Supplementary Figure 1. The binding picture of PbC₂O₄ on FAPbI₃ in DFT.

Supplementary Figure 8. HAADF-STEM of (a) control and (c) LOS perovskite. (b) and (d) are the enlargement of the area within the yellow square in (a) and (b), respectively. The inter-planar spacing of 0.38 nm (b) and 0.32 nm (d) matches the (111) and (002) reflection of the cubic α -FAPbI₃ perovskite phase, respectively.

Revision in manuscript:

We performed DFT simulation for the interface binding between FAPbI₃ and PbC₂O₄, as shown in Supplementary Fig. 1.

The high-angle annular dark field scanning transmission electron microscopy (HAADF-STEM) was used to probe the interface between PbC₂O₄ and FAPbI₃ and it can be clearly observed that there is a crystalline FAPbI₃ region and an amorphous PbC₂O₄ region (Supplementary Fig. 8).

(2) *According to the calculation, reducing the I vacancies is the most effective strategy for suppressing the α - δ phase transition. However, many substances have been reported to significantly reduce the I vacancies and what's the advantage of PbC₂O₄?*

Response: Thanks for your valuable comments. Many materials have been really reported to reduce the I vacancies on perovskite surface. For example, the use of an organic halide salt, such as phenethylammonium iodide (PEAI), alkylammonium bromide and n-Octylammonium Iodide (OAI) *et al.*, on perovskite films for surface passivation was a generally strategy to reduce the I vacancies. In comparison with organic halide salt, there are no I ions in PbC₂O₄ that will move during device operation. Namely, the halogen ions in the organic halide salt passivator will still move and form V_I while the large C₂O₄²⁻ oxalate with strong chelation in PbC₂O₄ is much stable. The steric hindrance of PbC₂O₄ can also impede the movement of mobile I ions from perovskite layer. Besides, PbC₂O₄ is chemically stable and insoluble in water that can protect the perovskite surface from the moisture corrosion. The solubility of PbC₂O₄ in water is merely 0.00065 g liter⁻¹, which is even lower than that of PbSO₄. However, most reported organic halide salt passivator, such as PEAi and OAI, are unstable under the moisture corrosion.

Revision in manuscript:

In comparison with the organic halide salt passivator, such as phenethylammonium iodide (PEAI) and n-Octylammonium Iodide (OAI) *et al*, PbC_2O_4 without mobile halogen ions is chemically stable and insoluble in water that can stabilize the perovskite surface well under the stimuli of moisture and electric field.

(3) *It is difficult to observe the crystal lattice from Figure 2c, d.*

Response: Thanks for your suggestions. We have observed the crystal lattice of control and LOS perovskite by HAADF-STEM as shown in Supplementary Fig. 8. Both in control and LOS films, the pronounced perovskite lattice can be observed. The crystalline region with an interplanar spacings of 0.36 nm and 0.32 nm was identified as the (111) and (200) of $\alpha\text{-FAPbI}_3$, respectively. (Nature Energy, 2021, 6, 419-428. Science 2023, 379, 288–294)

Supplementary Figure 8. HAADF-STEM of (a) control and (c) LOS perovskite. (b) and (d) are the magnification of the area within the yellow square in (a) and (c), respectively. The inter-planar spacing of 0.36 nm (b) and 0.32 nm (d)

matches the (111) and (200) reflection of the cubic α -FAPbI₃ perovskite phase, respectively.

(4) Are there any peaks corresponding to PbC₂O₄ in the XRD pattern of LOS perovskite films?

Response: Thanks for your suggestions. According to the XRD curves of Control-0 h and LOS-0 h in Fig. 3d, there is no any peaks corresponding to PbC₂O₄ (Fig. R1). We reason that the PbC₂O₄ is amorphous state as observed in the HAADF-STM images (Supplementary Fig. 8).

Figure R1. XRD patterns of the control and LOS films.

Reviewer #3 (Remarks to the Author):

This paper reported the pathway of α -to- δ phase and the effect of various defects on the phase-transition degradation of FAPbI₃ perovskite. They found that among the defects, the I vacancies easily generated on the surface and decreased the energy barrier of phase-transition, thus they are most likely to induce the α -FAPbI₃ degradation. Followed the DFT theoretical framework, a water-insoluble and stable lead oxalate was designed to stabilize the α -phase due to reduce the I vacancies and I migration. The resultant devices show very high efficiency and robust soaking stability. It is worthy publication in Nature Communications. Before it could be accepted, several points need to be clarified:

General response: We greatly appreciate the reviewer's positive and constructive comments and we have made the one-by-one responses to your questions raised. Please see the following contents.

(1) *There is lacking of a clear description on the interaction between H₂C₂O₄ and FAPbI₃. How about the uniformity of PbC₂O₄ layer on the perovskite? The PL mappings of the control and target films are also indispensable.*

Response: Thanks for your suggestions. H₂C₂O₄ has been generally used to combine metal ions to form precipitation in other fields. In our work, we have demonstrated the reaction between the oxalic acid and FAPbI₃ perovskite. After mixing oxalic acid and FAPbI₃ solution, a large amount of white-colored sediment was yielded which were identified as pure PbC₂O₄ materials through X-ray diffraction (XRD) patterns (Supplementary Figs. 2 and 3). Here, we give a description on the interaction as the following chemical equations: $\text{H}_2\text{C}_2\text{O}_4 + \text{FAPbI}_3 = \text{PbC}_2\text{O}_4 + \text{FAI} + \text{HI}$. We believe that the FAI can be dissolved in the IPA and thus removed during the preparation process.

Revision in manuscript:

Thus, a thin and compact PbC_2O_4 can be rapidly generated on the surface of the perovskite films through immersing the perovskite film into the oxalic acid solution or spin-coating the oxalic acid solution on perovskite, according to the following chemical equations: $\text{H}_2\text{C}_2\text{O}_4 + \text{FAPbI}_3 = \text{PbC}_2\text{O}_4 + \text{FAI} + \text{HI}$. The FAI can be dissolved in the IPA and thus removed during the preparation process.

As for the uniformity of PbC_2O_4 , we have observed some small white particles (PbC_2O_4) uniformly distributed on the perovskite surface in Fig. 2b. We further surveyed the perovskite film with a high concentration oxalic acid treatment by SEM (Fig. R2). The results confirmed that there is a uniform reaction layer on the perovskite surface. In addition, we measured the PL mapping of control and target films, as shown in Supplementary Fig.11. The PL intensity of LOS films was higher than that of control film in the whole $10 \times 10 \mu\text{m}^2$ region, further clarifying that the generated PbC_2O_4 uniformly passivate the perovskite film.

Figure 2b. Top-view SEM image of LOS perovskite film.

Figure R2. SEM image of LOS perovskite films treated with a higher concentration of oxalic acid of 4 mg mL⁻¹.

Supplementary Figure 11. PL mappings of (a) control and (b) LOS films in 10 × 10 μm² region.

Revision in manuscript:

In the PL mapping images, the intensity of LOS film in the whole 10 × 10 μm² region was higher than that of control film (Supplementary Fig.11), clarifying that the PbC₂O₄ uniformly passivates the perovskite film.

(2) *The effect of bare IPA on the device performance should be studied to excluding the role of IPA.*

Response: Thanks for your suggestions. We have compared the performance for the devices with and without IPA treatment. 40 μL IPA was spin-coated on perovskite surface at 4000 rpm for 30 s. As shown in Fig. R3, there is no obvious effect of IPA on the device performance. It is well known that IPA is a

general passivation solvent, which is not destructive to the perovskite layer in short time (Nature Photonics, 2019, 13, 460-466).

Figure R3. *J*-*V* curves of PSCs without and with bare IPA treated.

(3) *The IPA was rarely used as an anti-solvent to achieve high efficiency PSC. Most used diethyl ether or CB in one-step method. More detail about this method is needed.*

Response: Thanks for your suggestions. In fact, the IPA has been used as anti-solvent to prepare the perovskite film (A general approach to high-efficiency perovskite solar cells by any antisolvent, *Nature communications*, 2021, 12, 1878). It demonstrated that IPA antisolvents result in polycrystalline perovskite films with a remarkably high degree of preferred orientation. Inspired by this work, we further developed the type of method through adding a little of MAI in the IPA to achieve better crystalline quality perovskite film. We have also compared performance of PSCs based on different anti-solvents. Through XRD and SEM characterization of perovskite layers fabricated by different antisolvents, we identified that IPA can achieve better crystalline quality of perovskite films (Supplementary Figs. 15 and 16).

Revision in manuscript:

The IPA was selected as an anti-solvent since it resulted in better crystal quality and higher preferred orientation in perovskite films in comparison with the diethyl ether (DE) or chlorobenzene (CB)³⁷ (Supplementary Figs. 15 and 16).

Supplementary Figure 15. XRD of the perovskite films with IPA, CB and DE anti-solvent, respectively.

Supplementary Figure 16. SEM images of the perovskite films with (a) IPA, (b) CB and (c) DE anti-solvent, respectively.

(4) Besides V_i , how about the changes of the other defect formation energies after introducing the LOS?

Response: Thanks for your suggestions. As shown in Supplementary Table 2, we have added the detailed changes of other defects formation energies after introducing the LOS. For all the type I and III defects which are harmful to phase-stability, their formation energies are improved after introducing LOS. Thus, the results confirm the role of LOS in suppressing the generation of the detrimental defects.

Revision in manuscript:

The defect formation energies after introducing the LOS were given in Supplementary Table 2.

Supplementary Table 2. The formation energies of native point defects on the surface after introducing the PbC_2O_4

Type	Type I			Type II			Type III		
Defect	V_{Pb}	V_{FA}	I_{Pb}	Pb_i	Pb_{FA}	I_{FA}	V_i	I_i	Pb_i
Formation energy (eV)	2.97	2.76	3.73	1.76	1.73	3.38	1.53	1.92	1.46

(5) *Some English need correction. Such as Line 38 “various composition..... developed.” Line 72 “which are easily generated”, Line 168 “the LOS perovskite exhibited to three chemically”.*

Response: Thanks very much for the corrections, the modified contents are as follows:

Line 38: Various complex perovskite compositions have been developed to prevent the formation of the δ -phase.

Line 72: It is predicted that the I vacancies, which easily generated during the film preparation process and device operation, are most likely to trigger the α -to- δ phase-transition.

Line 168 (Revision in Line 180): In the O 1s spectra, the LOS perovskite exhibits three chemically distinct O 1s peaks at 531.7 eV, 532.5 eV, and 533.4 eV corresponding to the O atoms of deprotonated COO, C=O and OH components of the intact oxalate group.

REVIEWERS' COMMENTS

Reviewer #1 (Remarks to the Author):

The authors have well addressed the issues from the reviewers. The manuscript is now accepted for publication as is.

Reviewer #2 (Remarks to the Author):

The manuscript has been properly amended and I recommend its publication.

Reviewer #3 (Remarks to the Author):

The author has addressed all comments. It can be accepted as is.